# Epigenome-wide association study of human frontal cortex identifies differential methylation in Lewy body pathology

Lasse Pihlstrøm [1] ✉, Gemma Shireby[2], Hanneke Geut [3,4], Sandra Pilar Henriksen[1], Annemieke J. M. Rozemuller[5], Jon-Anders Tunold[1,6], Eilis Hannon [2], Paul Francis [2], Alan J. Thomas[7], Seth Love[8], Jonathan Mill [2], Wilma D. J. van de Berg[3] & Mathias Toft[1,6]

Parkinson's disease (PD) and dementia with Lewy bodies (DLB) are closely related progressive disorders with no available disease-modifying therapy, neuropathologically characterized by intraneuronal aggregates of misfolded α-synuclein. To explore the role of DNA methylation changes in PD and DLB pathogenesis, we performed an epigenome-wide association study (EWAS) of 322 postmortem frontal cortex samples and replicated results in an independent set of 200 donors. We report novel differentially methylated replicating loci associated with Braak Lewy body stage near *TMCC2*, *SFMBT2*, *AKAP6* and *PHYHIP*. Differentially methylated probes were independent of known PD genetic risk alleles. Meta-analysis provided suggestive evidence for a differentially methylated locus within the chromosomal region affected by the PD-associated 22q11.2 deletion. Our findings elucidate novel disease pathways in PD and DLB and generate hypotheses for future molecular studies of Lewy body pathology.

Parkinson's disease (PD) and dementia with Lewy bodies (DLB) are progressive and debilitating neurodegenerative disorders with complex aetiology and overlapping clinical and neuropathological features. These disorders make up the second most common cause of neurodegeneration and dementia after Alzheimer's disease (AD), and as their prevalence increases with longer life expectancy, the current estimate of 6 million PD patients worldwide is anticipated to more than double by 2040[1,2]. Current treatment for PD and DLB is merely symptomatic, and improved understanding of the molecular disease mechanisms is urgently needed in order to facilitate the development of targeted disease-modifying therapy. Over the last decade, large-scale genome-wide association studies (GWAS) have identified an increasing number of common genetic risk loci for PD[3], and more recently also for DLB[4,5], providing novel insights into pathogenesis. In contrast, previous studies have only explored to a limited degree how disease susceptibility and progression may be shaped by epigenetics. Epigenetic processes contribute to variability in complex traits such as aging and disease[6,7] and involve mechanisms of gene regulation that are partially dynamic over time, cell-type specific and influenced by both genetic and environmental factors.

DNA methylation at CpG dinucleotides is the most frequently studied epigenetic mechanism in complex disease. Previous EWAS analyses of AD-associated neuropathology have identified replicable and consistent patterns of differentially methylated CpGs across the

[1]Department of Neurology, Oslo University Hospital, Oslo, Norway. [2]University of Exeter Medical School, College of Medicine and Health, University of Exeter, Exeter, UK. [3]Amsterdam UMC, Vrije Universiteit, Department of Anatomy and Neurosciences, Amsterdam Neuroscience, Amsterdam, The Netherlands. [4]Netherlands Brain Bank, Netherlands Institute of Neurosciences, Amsterdam, The Netherlands. [5]Amsterdam UMC, Vrije Universiteit, Department of Pathology, Amsterdam Neuroscience, Amsterdam, The Netherlands. [6]Institute of Clinical Medicine, University of Oslo, Oslo, Norway. [7]Translational and Clinical Research Institute, Newcastle University, Newcastle Upon Tyne, UK. [8]Dementia Research Group, Bristol Medical School, University of Bristol, Bristol, UK. ✉e-mail: lasse.pihlstrom@medisin.uio.no

genome in postmortem human brain tissue, contributing important insights to disease mechanisms[8–11]. In contrast, similar efforts in PD or DLB have thus far only been reported for very limited sample sizes[12–14], although larger EWAS in PD have been performed on whole blood[15–17]. A recent study investigated frontal cortex neurons isolated by a flow cytometry-based approach and detected widespread differences in DNA methylation between 57 PD patients and 48 controls, with replication in a smaller dataset, highlighting dysregulation of *TET2* in particular[18].

Lewy bodies and Lewy neurites are intraneuronal aggregates composed mainly of misfolded α-synuclein protein and constitute the neuropathological hallmark lesions of both PD and DLB[19]. According to the staging system proposed by Braak[20], Lewy pathology in PD spreads in a predictable pattern over the course of the disease from the olfactory bulb and brainstem (Braak Lewy body stages 1–3) in a caudal-to-rostral manner, next to limbic regions (Braak Lewy body stage 4), reaching the neocortex in the last stages (Braak Lewy body stages 5 and 6). Only the retrospective information about clinical symptom progression distinguishes late-stage PD from DLB, where cortical Lewy pathology and associated cognitive symptoms are already present early in the disease course[21]. Of note, cellular and proteomic studies have demonstrated that neuronal stress and dysfunction are evident in the cortex also at the very early stages of PD-related pathology, long before Lewy bodies appear in the same brain region[22]. Previous transcriptome studies on post-mortem human brain tissue across Braak Lewy body stages have also highlighted early changes[23]. We hypothesized that epigenetic dysregulation contributes to the common molecular pathogenesis of Lewy body disorders, and that an increasing degree of disease-related changes should be detectable in frontal cortex as the disease progresses into more advanced stages.

To dissect the role of DNA methylation changes in PD and DLB, we performed a methylome-wide association study of 322 postmortem frontal cortex samples with Braak Lewy body stages 0–6 and replicated our findings in an independent dataset from 200 donors. The size and direction of effects for the most associated probes were highly correlated across the two datasets, providing a strong indication of consistent results. We report 4 replicating CpGs significantly associated with Braak stage, explore these loci in relation to known genetic risk factors and publicly available data and provide a benchmark for further epigenetic studies of post-mortem human brain tissue in Lewy body disorders.

## Results

### Differentially methylated CpGs across Braak Lewy body stages

For the discovery stage of our analysis, we used superior frontal gyrus grey matter tissue samples from the Netherlands Brain Bank (NBB) and Normal Aging Brain Collection, Amsterdam (NABCA) with available data on Lewy body Braak stage. We included samples from controls without any records of neurological or psychiatric disease during life ($n = 73$), donors without clinical neurological symptoms but with incidental Lewy body disease at autopsy (iLBD) ($n = 29$), clinically diagnosed and pathologically confirmed PD patients ($n = 139$) and DLB patients ($n = 81$) (Supplementary Table 1, Supplementary Figure 1). Aiming to focus our analysis on this spectrum of Lewy body disease, we excluded donors with pathologically confirmed AD either alone or in addition to one of these diagnoses. Samples were screened for monogenic causes of parkinsonism using the Illumina NeuroChip and no definitely or probably pathogenic variants were discovered in *SNCA*, *LRRK2*, *VPS35* or other relevant neurodegenerative genes covered by the array[24]. Genome-wide DNA methylation was assessed using the Illumina Infinium MethylationEPIC array. Following stringent quality control and filtering (see Methods), data on 583,192 CpG probes in 322 individuals were included in statistical analyses. We used a published algorithm and reference data to estimate the proportion of NeuN positive neurons in each tissue sample and included this variable as a

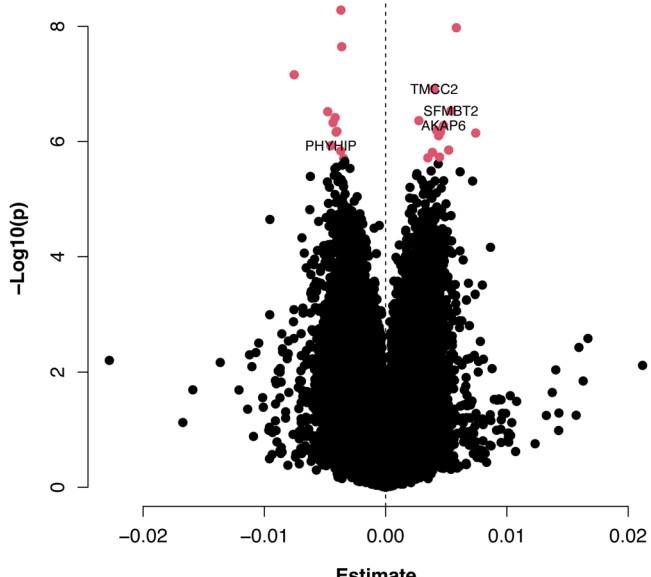

**Fig. 1 | Volcano plot of discovery stage linear regression results.** The figure shows linear regression effect estimates on the *x*-axis and −log(10) *p*-values on the *y*-axis for all probes in the NBB discovery analysis ($n = 322$ biologically independent samples). Red dots represent probes significantly associated when adjusting for multiple testing using the Benjamini-Hochberg false discovery rate method (two-sided FDR < 0.05). The four probes replicating at two-sided $p < 0.05$ in the BDR replication analysis ($n = 200$ biologically independent samples) are highlighted by closest gene labels.

covariate (see Methods). We observed a non-significant trend towards lower estimated proportion of neurons in samples with higher Braak Lewy body stage (Pearson $r^2$ −0.065, $p = 0.24$).

Lewy pathology is by definition strongly correlated with PD and DLB (see Methods and Supplementary Figure 1). In line with previous EWAS studies of AD pathology[11], we defined Braak Lewy body stage as the primary phenotype of interest under the assumption that this variable captures important shared aspects of pathogenesis in Lewy body disorders, thus not including diagnosis as a covariate in the primary statistical model. In a linear model adjusting for age at death, sex, postmortem interval, experiment plate, and the first 3 surrogate variables (see Methods), 24 CpG probes were associated with Braak α-synuclein stage at FDR < 0.05 in the discovery stage (Fig. 1, Supplementary Data 1). Next, we used equivalent inclusion criteria and analysis pipeline to analyse a replication dataset from 200 donors in the UK Brains for Dementia (BDR) cohort with DNA methylation data available[25]. Of note, the distribution of individuals across Braak Lewy body stages 0–6 was different across the two datasets, with a higher proportion of neuropathologically healthy donors in the replication cohort (68% vs 23% Braak stage 0) (Supplementary Table 1). We observed that DNA methylation differences across Braak Lewy body stages were strongly correlated across the two datasets for the 24 FDR-significant sites identified in the discovery phase (Pearson $r^2$ 0.61, $p = 0.0015$, 87.5% concordant direction, binomial sign test $p = 0.00028$) (Fig. 2). Based on this clear indication that the general pattern of differential methylation is reproduced in the BDR dataset, we considered it justified to interpret a two-sided $p < 0.05$ as evidence of replication, although we appreciate that a more conservative threshold adjusting for 24 independent tests would have been ideal. Associations for 4 DNA methylation sites were replicated at $p < 0.05$ in the BDR dataset including cg07107199 (annotated to *TMCC2*) (Fig. 3), cg14511218 (not annotated to any coding gene, located downstream of *SFMBT2*), cg09985192 (annotated to *AKAP6*) and cg04011470 (annotated to *PHYHIP*) (Table 1).

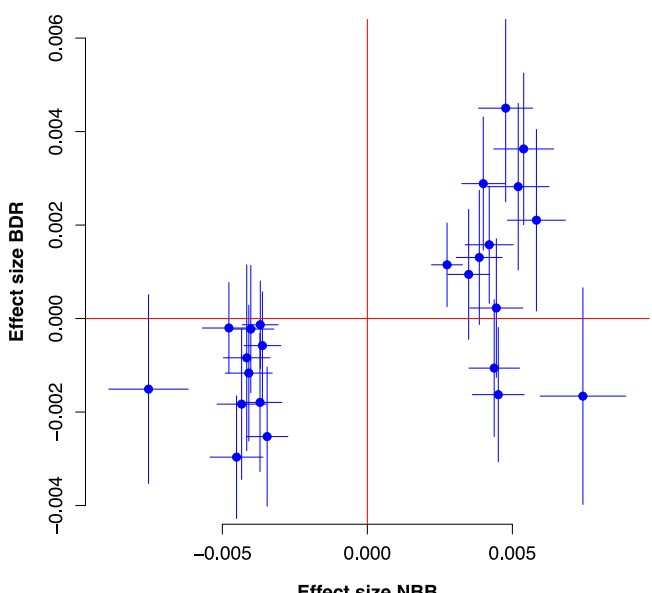

**Fig. 2 | Correlation between effect size in discovery and replication data for top probes.** The figure shows the coefficients ± SE from linear regression for the 24 probes passing a false discovery rate threshold (two-sided FDR < 0.05) in the discovery stage. A strong correlation between the discovery (NBB, Netherlands Brain Bank, $n$ = 322 biologically independent samples) and replication (BDR, Brains for Dementia Research, $n$ = 200 biologically independent samples) analyses indicates consistent results (Pearson $r^2$ 0.61, two-sided $p$ = 0.0015, 87.5% concordant direction, binomial sign test two-sided $p$ = 0.00028). Thin blue lines represent standard errors.

In addition to the two-stage discovery and replication approach, we also meta-analysed the 564,387 probes passing QC in both datasets. In a fixed-effect inverse variance meta-analysis, 35 probes were significant at FDR < 0.05 and 14 of these were associated at $p$ < 0.05 in both datasets with the same direction of effect (Supplementary Data 2). As expected, the 4 significant DNA methylation sites identified by the two-stage approach showed the strongest association signals also in meta-analysis. In addition, cg03318382 (not annotated to any coding gene, located upstream of *SEPTIN5*) emerged with the fifth strongest p-value in the meta-analysis, despite not being FDR-significant in the discovery stage alone. In meta-analysis, the three most significant probes also passed the more conservative Bonferroni-corrected threshold ($p < 8.9 \times 10^{-8}$, correcting for 564,387 tests). Quantile-quantile and Manhattan plots are shown in Supplementary Figures 2 and 3.

We did not expect genome-wide significant results in the smaller BDR dataset alone. We note however, that a set of probes showing the strongest trend towards association with Braak stage were all annotated to *CYP2E1* (top probe cg05194426; $p$-value = $2.65 \times 10^{-6}$, effect estimate = 0.0209, SE = 0.0043) (Supplementary Figure 3b). This observation is in line with previous smaller PD postmortem brain studies[13,26], yet we found no similar signal in the NBB dataset (cg05194426; $p$-value = 0.23, effect estimate = −0.004, SE = 0.0033) (Supplementary Figure 3a).

### Alternative statistical models
The inflation of test statistics in the discovery analysis was estimated at $\lambda$ = 1.177, yet fell to $\lambda$ = 1.077 using the bacon method developed specifically for epigenome- and transcriptome-wide studies[27]. This is well in line with recent large EWAS studies on AD neuropathology[10], as experience indicates that the low λ values typically seen in high-quality GWAS cannot be expected in methylation studies. A powerful, yet highly conservative approach developed to eliminate potential unobserved confounding is MOA (mixed linear model-based omic association), where the random effect of total genome-wide DNA methylation captures the correlation structure between probes and directly

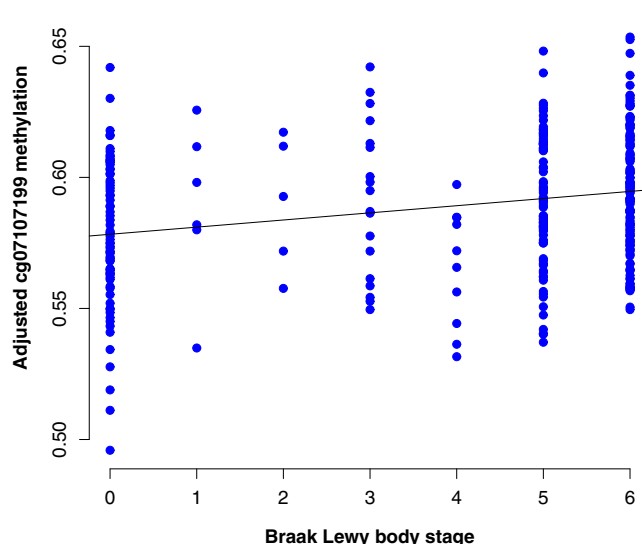

**Fig. 3 | Adjusted methylation values for the significant probe cg07107199 across Braak stages.** The figure shows individual methylation values in the discovery dataset ($n$ = 322 biologically independent samples), adjusted for covariates as described in the Methods section. A regression line is shown indicating increasing methylation levels with higher Braak α-synuclein stage (Intercept = 0.578, coefficient = 0.0027).

controls for the genomic inflation[28]. Comparing results obtained using MOA as implemented in the OSCA software (see Methods) to our linear regression model, we found that effect estimates were strongly correlated (Pearson $r^2$ 0.91) and effect directions all concordant (Supplementary Figure 4a), yet no probe reached genome-wide significance in MOA analysis (Supplementary Data 3). In the replication dataset, cg14511218 and cg04011470 still passed $p$ < 0.05 in the MOA analysis.

In another alternative model we assessed association with Braak Lewy body stage as a binarized variable, comparing stage 0-3 versus 4–6. As expected, this model generated highly similar results in terms of top probes and effect sizes (Supplementary Figure 4b), but was less powerful, with no probe reaching genome-wide significance at FDR < 0.5 (Supplementary Data 4). Finally, we ran linear regression with Braak Lewy body stage while adjusting for neuropathological diagnosis in addition to other covariates. This analysis showed that for the top probes, a weaker association remained ($p$ < 0.05) with consistent direction as the primary analysis, indicating that the strongest association signals are likely driven by differences in Lewy body stage partly across and partly within diagnostic groups (Supplementary Data 5, Supplementary Figure 4c).

### Phenotypic variance attributable to all probes
We used the omics restricted likelihood method (OREML) implemented in the OSCA software package to estimate the phenotypic variance for Braak Lewy body attributable to all probes in the dataset[28]. Including, sex, age at death, postmortem interval and experiment plate as covariates in the model, the proportion of variance captured by the omics relationship matrix ($V_{ORM}/V_p$) in the discovery dataset was 0.81 (SE = 0.18, $p$-value = $2.2 \times 10^{-7}$). Including the first three surrogate variables yielded a similar estimate ($V_{ORM}/V_p$ = 0.81, SE = 0.19, $p$-value = $1.1 \times 10^{-4}$).

### Exploring the genomic context of differentially methylated CpGs
Sites profiled by the MethylationEPIC array primarily map to promoters and enhancers in the human genome where variability in CpG

**Table 1 | Replicating CpG probes associated with Braak Lewy body stage**

| Probe | Position (b37) | Closest coding gene | Effect (SE) NBB | P NBB | FDR NBB | Effect (SE) BDR | P BDR | P meta | Genomic context |
|---|---|---|---|---|---|---|---|---|---|
| cg07107199 | chr1:205215911 | *TMCC2* | 0.0040 (0.0007) | 1.2e-07 | 0.014 | 0.0029 (0.0014) | 0.043 | 1.1e-8 | Enhancer Neun, Olig2 and LHX2 |
| cg14511218 | chr10:6962843 | *SFMBT2* | 0.0054 (0.0010) | 2.9e-07 | 0.025 | 0.0045 (0.0020) | 0.025 | 2.0e-08 | Enhancer PU1 and LHX2 |
| cg09985192 | chr14:32797255 | *AKAP6* | 0.0048 (0.0009) | 5.4e-07 | 0.026 | 0.0045 (0.0020) | 0.025 | 2.5e-08 | Promoter NeuN and Olig2 |
| cg04011470 | chr8:22079330 | *PHYHIP* | −0.0045 (0.0009) | 1.2e-06 | 0.039 | −0.0030 (0.0013) | 0.024 | 9.2e-08 | Enhancer NeuN |

Effect estimates are coefficients from linear regression, corresponding to the change in methylation beta value for each unit of increase in Braak stage. Genomic context refers to publicly available data from Nott et al.[34] mapping cell specific promoters and enhancers in frontal cortex for neurons (NeuN), astrocytes (LHX2), oligodendrocytes (Olig2) and microglia (PU1).

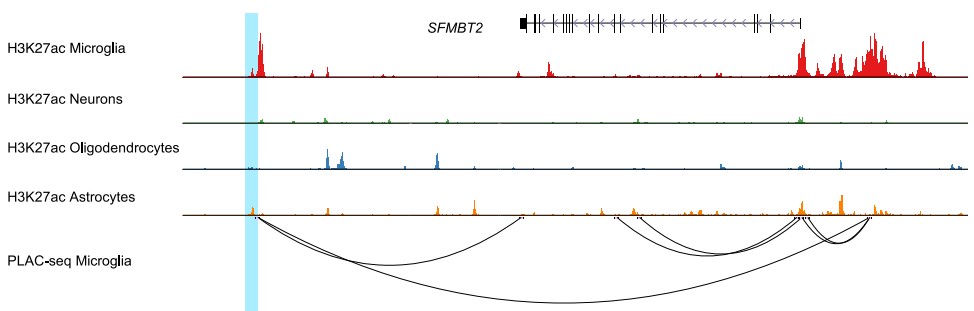

**Fig. 4 | The brain-specific genomic context of cg14511218 and relation to *SFMBT2*.** The figure shows chromosome 10, position 6,000,000–7,600,000 (b37), with the location of cg14511218 highlighted. Data from Nott et al.[34] show histone acetylation associated with active enhancers and promoters (H3K27ac) across different cell types, and proximity ligation-assisted chromatin immunoprecipitation sequencing (PLAC-seq) demonstrating enhancer-promoter contact in microglia.

methylation is likely to affect the functional regulation of nearby genes. To potentially nominate genes and cell-types implicated by our Braak Lewy body stage-associated CpGs, we assessed the overlap between probe positions and genomic annotations from a recent publication characterizing the noncoding regulatory regions and enhancer-promoter interactome of major cell types in the human cortex[29]. In this previous work, active promoters and enhancers were identified by assay for transposase-accessible chromatin sequencing (ATAC-seq) and chromatin immunoprecipitation sequencing (ChIP-seq) for histone modifications H3K27ac and H3K4me3. Table 1 includes a summary of probe position overlap with promoters and enhancers in neurons, microglia, astrocytes, and oligodendrocytes.

The strongest associated replicating probe, cg07107199 maps to an intron of *TMCC2* on chromosome 1, overlapping with a predicted enhancer in both neurons, oligodendrocytes, and astrocytes, and only 164 base pairs upstream of the transcription start site of a 4 exon isoform (ENST00000481950.2) of the *TMCC2* gene.

Cg14511218, is located ~240 kb downstream of *SFMBT2* on chromosome 8, where there is evidence of enhancer activity in microglia and astrocytes, but most pronounced in microglia (Fig. 4). Data from proximity ligation-assisted chromatin immunoprecipitation sequencing (PLAC-seq) in cortical microglia indicate that the enhancer makes its strongest interactions with the *SFMBT2* promoter[29]. Further evidence of the relevance of microglia to the cg14511218 association comes from the inclusion of *SFMBT2* in the human brain "microglial signature" published by Gosselin et al., comprising 881 transcripts showing minimum 10-fold increased expression in microglia relative to cortex tissue[30]. *SFMBT2* is also among the microglial genes highlighted as being differentially expressed in PD relative to control brains in a previous study[31].

Cg09985192 maps to a CpG approximately 1 kb upstream of *AKAP6*, overlapping a predicted enhancer region in neurons and oligodendrocytes. Cg04011470 is located in the last of 4 coding exons of *PHYHIP*. Epigenetic marks indicate a neuronal enhancer at the locus, interacting with several other genes in the region, some of which are also considered candidate genes for the *BIN3* PD GWAS association signal.

The DNA methylation site nominated primarily from meta-analysis, cg03318382, is located -13 kb upstream of *SEPTIN5*, not overlapping with a promoter or enhancer for any brain cell type. We note, however, that its location on chr22:19691974 (GRCh37) falls within the ~3 Mb region typically affected in the 22q11.2 deletion syndrome, which has been implicated in a number of clinical phenotypes, including idiopathic PD[32]. We found no evidence for the presence of any 22q11.2 deletion in the NBB dataset based on analysis of runs of homozygosity in genotype data (see Methods).

### Analyses of known genetic risk loci and methylation-quantitative locus analysis

Of note, one of the significant CpG probes (cg04011470) is located less than 500 kb from rs2280104 near *BIN3*, a genome-wide significant PD risk locus[3]. Comparing DNA methylation and genotype data in our discovery sample, we found no association between cg04011470 and rs2280104, nor with any other single-nucleotide polymorphism (SNP) within a 1 Mb window around the probe position. Similarly, we found no significant methylation-quantitative loci (mQTL) for cg04011470 among 277 cis-SNPs in publicly available frontal cortex data summary statistics from the Brain xQTL server (*n* = 543)[33] nor data from a 2018 brain mQTL study (estimated effective *n* = 1160)[34].

Turning to other known risk loci for PD, we extracted results from all 26,743 CpG probes located within 500 kb of any significant GWAS SNP reported by Nalls et al., but cg04011470 remained the only association significant when correcting for multiple testing. Hypomethylation of the *SNCA* promoter region in PD has been reported in a

number of previous studies, both in brain and blood[35–37]. We therefore assessed CpG probes in the *SNCA* locus specifically, but found no significant CpGs when correcting for multiple testing of 109 probes within 500 kb of significant SNPs (minimum p-value 0.0035 for cg01035160).

Next, we assessed both NBB genotypes and publicly available data for cis-mQTLs within 500 kb of our remaining 3 Braak stage-associated CpGs, but found no SNPs significantly associated with methylation levels for any of the probes. These results suggest that the DNA methylation differences associated with Lewy body pathology do not appear to be mediated by underlying genetic variability.

## Discussion

We performed a methylome-wide association study across Braak Lewy body stages in postmortem human frontal cortex, and followed up our findings in an independent dataset, identifying 4 novel differentially methylated loci that replicated across sample series. Including a total of 522 participants, we analysed a larger sample than the few previously published EWAS studies of human brain tissue in Lewy body disorders[13,14,26]. Our study design mirrors a series of successful bulk cortex EWAS studies of AD neuropathology, where the number of significant CpG associations has increased from a handful in the first published studies[8,9] to several hundred in recent meta-analyses[10,11], providing novel insights into disease pathways and mechanisms. We demonstrate here that the same approach is feasible also for Lewy body pathology, the hallmark of PD and DLB.

Differentially methylated sites are located in promoters or enhancers, and interpretations linking association signals to specific genes should be made with caution in the absence of functional evidence. The strongest associated replicating probe is located in a predicted enhancer within an intronic region of *TMCC2*, encoding transmembrane and coiled-coil domains protein 2, which is primarily expressed in brain and blood and localizes predominantly to the endoplasmatic reticulum[38]. The protein has been shown to interact with apolipoprotein E (APOE) in an allele-specific manner, with evidence indicating that interaction between *TCMM2* and the AD-associated APOE ε4 allele may alter amyloid β production[39]. Our findings suggest the possibility that *TMCC2* may also have a role in α-synuclein pathology.

Based on genomic annotations from human cortex overlapping with the location of cg14511218, we raise the hypothesis that dysregulation of *SFMBT2* in microglia may be linked to Lewy body pathology. *SFMBT2* encodes Scm-like with four malignant brain tumour domains 2. Apart from being reported as a microglial signature gene[30] and differentially expressed in PD[31], its function in the human brain is largely unexplored. *AKAP6* upstream of cg09985192 encodes A-kinase anchoring protein 6, which is abundantly expressed in various brain regions and striated muscle. Genetic variability in the *AKAP6* locus has been associated with general cognitive function in a large GWAS meta-analysis, but the implicated molecular mechanism is unknown[40].

We noted with interest that the differentially methylated probe cg04011470 in *PHYHIP* was located less than 500 kb from the top associated SNP identified in a recent PD GWAS analysis. Studies of the association between common SNPs, gene expression, histone modifications and CpG methylation in the human brain have revealed that the different types of QTLs show a certain degree of overlap and are all enriched among disease-associated GWAS SNPs[33], raising the possibility that our finding could be linked to the same mechanism as the adjacent GWAS signal. However, in the same study, SNP effects on RNA expression were only mediated fully by epigenetic variation in 9% of loci, so simple one-to-one relationships remain the exception. Consequently, we are not surprised that we did not identify significant mQTLs for cg04011470 or any of the other differentially methylated CpGs, which is also in line with a recent large meta-analysis of AD neuropathology EWAS[11]. The lack of mQTLs is still compatible with a hypothesis that genetic and epigenetic association signals from the same genomic region may represent independent disease mechanisms converging on the same gene.

In meta-analysis, we identified 14 CpGs associated with Braak Lewy body stage with consistent direction of effect and $p < 0.05$ in both the NBB and the BDR datasets. This approach is less conservative than the standard two-stage discovery and replication method and should therefore be interpreted with caution. The strongest additional signal emerging in our meta-analysis was an association between Lewy pathology and hypermethylation of cg03318382, located within the ~3 Mb region typically affected in the 22q11.2 deletion syndrome. This deletion occurs in at least 1 in 4000 births and has been linked to a number of clinical phenotypes including DiGeorge syndrome, velo-cardiofacial syndrome and psychotic illness[41]. Early-onset parkinsonism reported as a clinical feature in known carriers of the 22q11.2 deletion sydrome[42] prompted further investigations confirming its role as a risk factor for PD, also predisposing to early onset[32,43]. It is not clear which genes within the region are important for PD pathogenesis, although *COMT*, encoding catechol-o-methyltransferase has been highlighted as an interesting candidate[32]. We found CpG hypermethylation upstream of *SEPTIN5*, which encodes septin 5. Septins are guanosine-5'-triphosphate (GTP)-binding proteins that contribute to the regulation of synaptic vesicle trafficking and neurotransmitter release, and septin 5 has been shown to interact with parkin, a protein implicated in early-onset autosomal recessive PD[44]. Our findings highlight the possibility that epigenetic changes in the 22q11.2 region may be associated with disease risk in individuals with normal copy number due to dysregulation of one or more of the same genes that contribute to the 22q11.2 deletion syndrome.

Hypomethylation at the promoter in intron 1 of *SNCA* has been reported in postmorten human brain. One study showed significant differences in both substantia nigra and cortex[35], whereas another found significant change in substantia nigra only[36], with similar methylation levels in patients and controls in putamen and the anterior cingulate. Furthermore, *SNCA* hypomethylation has been demonstrated in whole blood and cultured mononuclear cells, also with an mQTL association to susceptibility SNPs for PD and DLB in brain and blood[37,45], but the methodologies used have been different from the present study. A number of factors, including choice of brain region, array design and adjustment for cell composition may have contributed to our inability to replicate this signal.

In the BDR dataset, we note a trend towards association with Braak stage for a number of probes annotated to *CYP2E1*, yet there was no similar signal in the larger NBB data. The association is not FDR-significant in BDR data, but the region stands out from the rest of the results, as is apparent in the Manhattan plot (Supplementary Figure 3b). In line with this observation, an early genome-wide DNA methylation scan using in postmortem brain samples from 6 PD patients and 6 control donors highlighted differential methylation at the *CYP2E1* locus[13]. Furthermore, while our manuscript was under review, a follow-up study in 14 PD samples and 10 controls was published, providing further evidence for hypomethylation of *CYP2E1* in PD cortex, albeit not passing a significance level adjusted for multiple testing[26]. In the same study, hypomethylation correlated with increased expression of the cytochrome P450 2E1 protein, but not with recorded dose of L-dopa medication during life. We have no explanation for the discrepant *CYP2E1* results across our two datasets, and further research is needed to clarify the role of this locus in Lewy body disorders. We note that CYP regulation is sensitive to environmental exposures, many of which may potentially differ for donors from the Netherlands and the UK.

Our study has several limitations. Differentiating causes from effects is a constant challenge in epigenetic studies of complex disease, and particularly difficult for brain disorders, where longitudinal sampling is impossible for the main tissue of interest[46]. Comparing

iLBD to control donors without evidence of Lewy body pathology in independent samples could potentially clarify if methylation changes occur at a preclinical stage, but functional experiments will likely be required to establish causality. The standard sodium bisulfite conversion method used in MethylationEPIC array analysis does not capture variability in hydroxymethylation, an intermediary DNA modification state between methylated and unmethylated cytosine that is enriched in the brain and serve specific regulatory functions[47]. Parallel profiling of methylation and hydroxymethylation has been shown to improve the resolution and interpretation of epigenetic studies in AD[48]. Depending on the availability of tissue and resources, we investigated only superior frontal cortex, where we ideally would have assessed tissue from multiple brain regions. Experience from AD EWAS indicate some regional variation in differential DNA methylation, yet substantial overlap between significant CpGs across different parts of the cerebral cortex[11]. Substantia nigra is particularly vulnerable to Lewy pathology and pivotal in PD pathogenesis, yet postmortem methylation studies of this region is of limited value due to the profound neuronal loss. We used a published algorithm to estimate cell composition from methylation profiles and adjust for this variable in our analysis. The reference dataset only distinguishes NeuN positive neurons from non-neuronal cells, however, and even with good bioinformatic corrections, epigenetic studies of bulk tissue will never obtain a resolution comparable to investigations of pure cell populations.

A recent study isolated frontal cortex neurons by a flow cytometry-based approach and reported widespread methylation differences between 57 PD patients and 48 controls, with replication in a smaller dataset[18]. In particular, the authors highlighted dysregulation of *TET2*, involving a pattern of promoter hypomethylation and enhancer hypermethylation. This study assessed methylation by bisulfite padlock probe sequencing, so the results are not directly comparable to our MethylationEPIC array data. We did not detect significant association signals from the *TET2* locus, but we note that only a subset of CpGs were differentially methylated in the original study, and our data did not precisely cover the same positions. We acknowledge that methylation studies in sorted cell populations have many advantages over our design, as less noisy data are both more powerful and easier to interpret. However, as bulk tissue approaches like ours remain far more scalable to large sample sizes, we anticipate that both cell type specific and bulk brain tissue EWAS designs will have important roles to play in mapping the epigenetics of Lewy body neuropathology. We also note that there is ample evidence to support a role also for non-neuronal cells in the pathogenesis of Lewy body disorders[49,50]. The large-scale bulk tissue design has the ability to potentially pick up strong signals driven by any cell type, as our exploration of overlap between differentially methylated CpGs and functional genomic annotation for major brain cell types also indicates.

Our primary analysis model assumed Braak Lewy body stage as a quantitative variable and did not take diagnosis into account, following the approach of previous major EWAS of AD neuropathology[8,9,11]. We acknowledge as a major limitation, however, that our dataset is not powered to dissect with any accuracy to what extent associations are driven directly by Braak Lewy body stage versus being an indirect effect of PD and DLB diagnosis. Furthermore, we also lacked data on environmental exposures that might be relevant, in particular smoking, which is known to affect DNA methylation and be associated with Lewy body disorders in epidemiological studies (see Methods).

Despite being larger than previous brain EWAS in PD and DLB, the sample size of our study was still modest, limiting our statistical power for hypothesis-free association analysis. We note that although the overall pattern of association of the top probes was highly similar, discovery phase results did not reach FDR adjusted significance in the alternative analysis approach based on the more conservative MOA method. It is biologically plausible, however, that Lewy body pathology has widespread effects on methylation causing some degree of inflation that does not result from confounding, which is also compatible with the high observed estimate of the proportion of variance attributable to all probes (0.81). Furthermore, we considered it justified to interpret a two-sided $p < 0.05$ in the BDR data as evidence of replication, yet we acknowledge that a significance threshold correcting for multiple testing also in the replication stage would have been more robust.

As the total number of samples have passed one thousand for meta-analyses of EWAS in AD neuropathology, a larger set of robustly associated CpGs can be utilized for enrichment analyses, linking the results collectively to molecular mechanisms and pathways. We hope to see a similar development in Lewy neuropathology EWAS through collaborative efforts in the years to come. Furthermore, cross-trait analyses are warranted to assess the degree of epigenetic overlap versus disease-specific methylation patterns across neurodegenerative disorders. Understanding causality and the temporal sequence of disease-linked processes is also of major importance and could potentially be addressed through studies in model systems as well as larger collections of postmortem samples in the very early stages of neuropathological change. Finally, more studies are needed to identify the genetic and environmental determinants driving disease-associated methylation changes in neurodegenerative disorders.

In summary, we performed the first well-powered two-stage EWAS of Lewy body neuropathology in postmortem human frontal cortex, providing evidence for significant methylation differences associated with Braak Lewy body stages and linking novel genomic loci to the pathology underlying PD and DLB. These findings generate hypotheses for further molecular studies and hold promise that future meta-analyses in larger sample sets will be as fruitful in Lewy neuropathology as in recent studies of AD-related changes. Complementing other genomic approaches such as GWAS and gene expression studies, investigations of DNA methylation may provide important contributions to our basic understanding of disease mechanisms and ultimately facilitate the development of disease-modifying therapy for PD and DLB.

## Methods
### Subjects and samples
Samples in the discovery dataset were obtained from the Netherlands Brain Bank (NBB, www.brainbank.nl) and Normal Aging Brain Collection, Amsterdam (NABCA)[51]. Written informed consent for the use of tissue samples and clinical information for research purposes was collected from the donors or their next of kin. The Medical Ethics Committee of the VU University Medical Centre, Amsterdam, approved all procedures of NBB and NABCA. The EWAS study was approved by the Regional Committee for Health and Medical Research Ethics, Norway. Standardized brain autopsies and neuropathological examinations were performed by experienced neuropathologist or neuroanatomist (AR and WB), including assessment of Lewy body-related α-synuclein pathology according to BrainNet Europe guidelines[52]. Clinical information was extracted from medical records provided by the NBB. PD diagnosis was based on the combination of clinical parkinsonism according to UK Parkinson's Disease Society Brain Bank[53] or Movement Disorders Society[54] criteria and moderate to severe loss of neuromelanin-containing neurons in the substantia nigra in association with Lewy pathology in at least the brainstem, with or without limbic and cortical brain regions[55]. Criteria for DLB were a clinical diagnosis of probable DLB according to the consensus criteria of the DLB Consortium[21] combined with presence of limbic-transitional or diffuse-neocortical Lewy pathology upon autopsy. Dementia was diagnosed clinically during life by a neurologist or geriatrician, or retrospectively based on neuropsychological test results showing disturbances in at least two core cognitive domains or Mini-Mental State Examination (MMSE) score <20. The distinction between DLB

and PD with dementia (PDD) (74 out of 139 PD patients) was made based on the "1-year rule", classifying dementia presenting before or within 1 year of parkinsonism onset as DLB[21]. Tissue slices of 50–100 mg macroscopically spanning all cortical layers, grey matter only, were cut from frozen frontal cortex tissue blocks, collected at autopsy and stored at −80 °C until further processing, in a cryostat, and DNA was extracted using the Qiagen AllPrep DNA/RNA/miRNA Universal Kit according to the manufacturer's instructions.

A subset of donors from the BDR DNA methylation dataset[25] were used to replicate specific DMPs from the discovery cohort. We used prefrontal cortex Illumina Infinium MethylationEPIC BeadChip data from 200 individuals. Briefly, the BDR cohort was established in 2008 and consists of a network of six dementia research centres across England and Wales (based at Bristol, Cardiff, King's College London, Manchester, Oxford, and Newcastle Universities) and five brain banks (the Cardiff brain donations were banked in London). BDR is approved as a Research Tissue Bank by the National Research Ethics Service. All participants have given informed consent. Details on recruitment and cohort setup have been thoroughly described in a previous publication[56]. Post-mortem brains underwent full neuropathological dissection, sampling, and characterization by experienced neuropathologists in each of the five network brain banks using a standardized BDR protocol which was based on the BrainNet Europe initiative[52]. Additional information on the BDR DNA methylation dataset can be found in Shireby et al.[25].

### Genotyping and mutation screening

All NBB samples were genotyped using the NeuroChip Consortium Array (Illumina, San Diego, CA USA)[24]. Quality control was carried out in PLINK 1.9[57]. Quality control included filtering of variants and individuals based on call rate (<0.95), Hardy-Weinberg equilibrium ($p < 0.000001$), relatedness (pi-hat > 0.125) and excess heterozygosity (>4 SD from mean) as well as sex-check and ancestry assessment based on principal component plots. Samples failing genotyping QC were excluded from all analyses. Genotypes were imputed using the Michigan Imputation Server[58] using default settings and reference data from the Haplotype Reference Consortium[59], and SNPs with an imputation $r^2 < 0.3$ were filtered out. The NeuroChip array has also been designed to allow for screening for known pathogenic mutations in relevant Mendelian neurodegenerative genes. We identified no carriers of definitely or probably pathogenic variants in *SNCA*, *LRRK2*, and *VPS35* in our discovery sample set. We used the "homozyg" function in plink 1.9 with window size 2.5 Mb and otherwise default parameters to screen for runs of homozygosity on chromosome 22, which could indicate the presence of a 22q11 microdeletion.

### DNA methylation analyses, data normalization and quality control

500 ng DNA from each sample was bisulfite treated and assessed using the Illumina Infinium MethylationEPIC BeadChip, assigning samples randomly to arrays. The NBB experiment included technical replicates for 30 samples, which were used for quality checks only and removed before statistical hypothesis testing. For both datasets, all data processing was performed using R 4.0.3. Raw signal intensity data were imported into R using the minfi v1.36.0[60] package. For rigorous quality control, we applied a series of checks and filtering steps taking advantage of the minfi and wateRmelon v.1.26.0[61] R packages. CpG sites with a beadcount <3 in 5% of samples or detection $p$-value > 0.05 in 1% of samples were filtered out using the pfilter function in the wateRmelon package. No samples had detection $p$-value > 5% in 5% of sites. Two samples in the NBB data and 19 samples in the BDR data were removed due to low median signal intensities as flagged by the minfi getQC function. No outliers were detected by the wateRmelon outlyx function or inspection of bisulfite conversion or other control probe metrics. Sex chromosome CpGs were used to estimate sample

sex and two samples failing sex-check were removed. We estimated the proportion of NeuN positive cells in each sample using reference data from flow sorted frontal cortex cell populations[62] as implemented in the minfi package. Two outlier samples with low NeuN proportions were removed from the NBB dataset. We evaluated different normalization methods, ultimately selecting the wateRmelon dasen method[61], which generated the minimum mean difference in beta value across technical replicate pairs in NBB data and has been successfully employed in previous AD EWAS with similar study design. The normalized MethylSet data object was mapped to the genome and probes on sex-chromosomes, probe sequences containing SNPs of any minor allele frequency in the MethylationEPIC annotation and previously reported cross-reactive probes[63] were filtered out. It has been shown that a considerable proportion of methylation array CpGs have large measurement errors, making them unsuitable for statistical association testing in complex disorders. Taking advantage of the technical replicates in the NBB dataset we used the CpGFilter v1.1 package to compute the intra-class correlation coefficient (ICC), which characterizes the relative contribution of the biological variability to the total variability for each probe[64]. The authors of CpGFilter recommend discarding all probes with ICC below the median. We chose, however, to only filter out the lowest quartile. Finally, the MethylSet data object was converted to beta values and the wateRmelon pwod function was used to filter out values lying more than four times the interquartile range from the mean, outliers assumed to result from rare SNP artifacts.

### Linear regression and mQTL analyses

There is no general consensus on whether to use beta or M-values for statistical analyses of methylation array data. We used beta values for our main analyses because this method has been favoured in previously published studies on AD neuropathology that inspired our work. However, we also repeated the statistical analyses with M-values, confirming that the main results obtained were highly similar (Supplementary Data 6, Supplementary Fig. 4d). We generated principal components (PC) and found that after the 5th PC, each PC explained <1% of the variation in the NBB dataset. We observed that NeuN positive cell type proportion and bisulfite conversion experiment plate were associated with PCs, and these were included as covariates in the final analyses in addition to sex, age at death and postmortem delay as recorded by the brain bank. We note that mean postmortem interval was shorter for PD and DLB patients than for controls in the NBB dataset and considerably longer overall in the BDR dataset. Although postmortem interval was not associated with any of the top 5 PCs in the NBB data, we chose to include it as a covariate, as evidence from rat experiments indicate that brain methylation is not stable in room temperature[65]. To further minimize the risk of unknown batch effects affecting the results, we estimated surrogate variables (SVs) using the sva v3.38.0 package[66]. In line with previous work, we evaluated models with different numbers of SVs and observed the effects on the test statistic distribution[11]. We found that including 3 SVs led the inflation measure λ to fall below 1.2 in the NBB analysis, without further improvement when additional SVs were added. Linear regression was performed using the limma package[67] with the following model in both discovery and replication:

Methylation $\beta$ ~ Braak Lewy body stage + sex + age at death + cell composition + experiment plate + postmortem interval + SV1-3.

We also explored alternative linear models including neuropathological diagnosis as a categorical variable (Methylation $\beta$ ~ Braak Lewy body stage + neuropathological diagnosis + sex + age at death + cell composition + experiment plate + postmortem interval + SV1-3) and defining Braak stage as a binary variable (0–2 versus 3–6).

In the main analysis we adjusted for multiple testing using the Benjamini-Hochberg false discovery rate method, which is recommended as default for microarray studies in the limma package.

Despite being somewhat less conservative than the Bonferroni method, we considered this to be justified given our two-stage design with independent replication. Fixed-effect meta-analysis was performed using the meta v5.2–0 package[68]. For visualization, we calculated the residuals from a linear regression model with covariates only, excluding Braak stage, adding these to the model intercept to generate covariate-adjusted methylation values. The same residuals were used as phenotype in linear regression mQTL analysis in PLINK 1.9. We applied Bonferroni-correction locus by locus to correct for multiple testing in the mQTL assessment. We downloaded data from the xQTL sverver[33] and the homepage of the summary-data-based Mendelian Randomization (SMR) tool[34] to assess mQTLs in publicly available data.

Smoking exposure during life would have been a relevant variable to include as a covariate but was not available in our brain bank data. Tools have been developed to estimate smoking exposure based on reference data from association studies of smoking and DNA methylation in whole blood, yet this kind of method has been shown to have limited value when applied to human postmortem brain data in an analysis using tissue from the nucleus accumbens, and the same study found no evidence of shared differentially methylated CpGs across blood and brain[69]. We nevertheless attempted to estimate smoking scores in the NBB data using the EpiSmokEr v0.1.0 package[70] with default parameters. All donors were classified as either former or current smokers, and quantitative scores were not associated with Braak Lewy body stage or neuropathological diagnosis. We therefore chose not to include any smoking estimate in the model.

### MOA and OREML analyses

We converted the final matrices of NBB and BDR methylation beta values from the DNA methylation data processing detailed above and converted to BOD-format for further analyses using the OSCA v0.46 software[28]. Probes with mean methylation beta values below 0.025 or above 0.975 where removed and probe values were adjusted for sex, age at death, cell composition, experiment plate and postmorten interval, but not surrogate variables. Mixed linear model-based omic association (MOA) was then applied to analyse association between Braak Lewy body stage and adjusted probes.

The same covariates were used when estimating the omics relationship matrix (ORM) using the OSCA software, followed by restriction maximum likelihood analysis (OREML) to estimate the phenotypic variance attributable to all probes.

### Reporting summary

Further information on research design is available in the Nature Research Reporting Summary linked to this article.

## Data availability

The methylation data generated in this study have been deposited in the Gene Expression Omnibus (GEO) database under accession codes GSE203332 (NBB) and GSE197305 (BDR). Additional data on NBB donors can be obtained from the Netherlands Neurogenetics Database (https://www.brainbank.nl/nnd-project/). Additional BDR donor data are available via the Dementias Platform UK (DPUK) data portal (https://portal.dementiasplatform.uk/). The processed summary statistics data are available at https://github.com/lpihlstrom/projects/tree/main/Lewy_pathology_EWAS/sumstats[71]. Other source data used to generate Fig. 3 and Supplementary Figure 4 are provided in the Source Data file. The publicly available mQTL data used in this study are available from the Brain xQTLServe database (http://mostafavilab.stat.ubc.ca/xqtl/) and the SMR software homepage (https://yanglab.westlake.edu.cn/software/smr/#mQTLsummarydata). Genotype imputation was performed using reference data available from the Haplotype Reference Consortium (http://www.haplotype-reference-consortium.org/). Source data are provided with this paper.

## Code availability

All analysis code used in this manuscript is available on GitHub at https://github.com/lpihlstrom/projects.[71]

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

## Acknowledgements

The study was funded by the Norwegian Health Association (grant 4799, L.P.) and the Research Council of Norway (grant 250597, M.T.). L.P. and M.T. also received additional funding from the South-Eastern Regional Health Authority, Norway, the Norwegian Parkinson Research Fund and Reberg's Legacy. W.D.J.v.d.B. received funding from the Dutch Parkinson association, Health Holland, and Rotary Club Aalsmeer-Uithoorn. The authors are grateful to the Netherlands Brain Bank and its funders for providing the samples that made the study possible. The Brains for Dementia Research cohort, including the generation of DNA methylation data, is jointly funded by the UK Alzheimer's Society and Alzheimer's Research UK.

## Author contributions

L.P., W.D.J.v.d.B., and M.T. designed the study. H.G., A.J.M.R. and W.D.J.v.d.B. provided clinical and neuropathological data for the NBB cohort. S.P.H. and L.P. performed wetlab work. L.P. performed analyses and drafted the manuscript. G.S. performed cohort-level analyses on the BDR data. J.-A.T. contributed to data handling and visualization. G.S., E.H., P.F., A.J.T., S.L. and J.M. organized and/or performed collection and generation of BDR data. J.M., W.D.J.v.d.B., and M.T. provided supervision. All authors contributed to critical revision of the manuscript.

## Competing interests

The authors declare no competing interests.
