## [Peer Review File · Nature Communications]

Epigenome-wide association study of human frontal cortex identifies differential methylation in Lewy body pathologyREVIEWER COMMENTS

Reviewer #1 (Remarks to the Author):

Summary

Pihlstrøm et al. generated DNA methylation (DNAm) data for a large cohort of 322 postmortem frontal cortex brain samples from iLB, PD, DLB and control subjects. They performed analysis on both discovery and replication datasets to nominate 4 significant CpGs. Given that few DNAm studies have been performed in large cohorts of PD samples so far, the study made a significant contribution to the field by generating a valuable resource. The Discussion section is well written. My major concern for the work is that some of the analytical methods used for analyses might not be appropriate. In particular, there is a lack of control for multiple comparison adjustments in the replication study. There was also a lack of discussion on why different models were used in discovery and replication datasets, as well as a lack of consideration for differentially methylated regions.

Major Comments

1. Appropriateness of linear models. In the discovery cohort, there are different groups of samples from subjects with iLBD, PD, and DLB, for which the association between DNAm and Braak stage can be heterogeneous. The linear model should also adjust for heterogeneities introduced by these different groups, in addition to age, sex and proportions of neurons, to avoid potential confounding effects.

2. Adjustment for cell-type proportions. Why were two different methods used for the analysis of discovery and replication cohort? For the discovery cohort, "We estimated the proportion of NeuN positive cells in each sample using reference data from flow-sorted frontal cortex cell populations as implemented in the minfi package."

For replication cohort, "Cell proportions were derived using an algorithm which classifies the cellular populations in the cortex into three proportions (neuronal enriched, oligodendrocyte enriched and microglial enriched) (Shireby et al., manuscript in preparation)." Why not use the same method for both datasets? especially given the second one is not yet published.

3. Why was the discovery and replication cohort analyzed using different models?

For discovery, "We used a published algorithm and reference data to estimate the proportion of NeuN positive neurons

in each tissue sample and included this variable as a covariate, as well as sex, age, postmortem interval, experiment plate, and the first three surrogate variables."

But for the replication cohort, "linear regression models against Braak Lewy body stages were run controlling for age, sex, experiment plate and derived cellular proportions"

Why weren't postmortem interval and SVs used for replication cohort?

4. Another major concern is the lack of correction for multiple comparisons. In the analysis of the replication cohort, 24 probes from the discovery cohort were studied, FDR or Bonferroni correction should be made.

5. In the section "Association between CpG methylation and preclinical Lewy body pathology", there was no multiple comparison correction for the 4 significant CpGs that were compared between neurologically healthy donors with iLBD (N = 29) and without iLBD (N = 73).

6. Please also provide FDRs associated with meta-analysis P-values in Supplementary Table 2.

7. I checked on the GitHub repository for scripts, but it is empty. The scripts should be made available before publication to allow assessment by reviewers on the reproducibility of the work.

8. I recommend changing supplementary files to excel format. It's much easier for other researchers to analyze and re-use the data presented in excel files than a pdf file.

9. The authors can also perform DMR analysis to detect differentially methylated regions in the datasets.

Minor comments

1. Please change age to age at death throughout the manuscript

2. For the quality control of DNAm datasets, did the authors look at bisulfite conversion rate?

3. In Supplementary Table 1, please clarify the difference between "DLB" and "LBD" in the legend.

4. For Supplementary Figure 1, I suggest using distinct colors to differentiate different stages

5. Figure 1 needs a better description. It might be better to use "effect size" rather than "beta values" throughout, as "beta values" can be confused with "methylation beta values". The legend also needs to indicate the estimated correlation and P-value.

6. In Supplementary Table 2 ",", should be "."

7. In the text throughout, please replace "*" with "x". For example, $p = 5.0 \times 10^{-8}$ should be 5.0×10^{-8}

8. The manuscript would benefit from editing from a writing service. For example, "Parkinson's disease (PD) and dementia with Lewy bodies (DLB) are closely related progressive disorders with no available causal therapy" – what does "causal therapy" mean?

Reviewer #2 (Remarks to the Author):

In this manuscript, Pihlstrom and colleagues describe analysis of DNA methylation in Lewy body pathology using data from 541 post-mortem frontal cortex samples from the Netherlands Brain Bank (discovery) and Normal Aging Brain Collection, Amsterdam (replication). This is a large sample for a brain based EWAS and is a significant new dataset in relation to understanding epigenetic changes involved in PD, LBD and DLB. The authors report 4 novel replicating differentially methylated probes, and a further locus from meta-analysis of the discovery and replication samples. These are notable findings, but I have several queries about the methods and interpretation of results.

Analysis method:

– Do I understand correctly that linear regression was used to analyse Braak stage as a continuously distributed response variable? Inclusion of an equation(s) summarizing the analysis approach would be helpful.

– Given that the distribution of Braak stage is skewed, and differs significantly between discovery and replication samples, did the authors compare their primary results to those based on analysis of ordinal (e.g., Braak 0/1 vs 2/3 vs 4-6) or binarized (e.g., Braak 0-2 to Braak 3-6) variables?

– Linear regression with covariates can result in significant inflation in EWAS due to unobserved confounding. Acknowledging the merits of the SV analysis, it would be reassuring if the results were robust using the mixed-linear-model-based MOMENT (and/or MOA) methods implemented in the OSCA package (<https://doi.org/10.1186/s13059-019-1718-z>), which have been shown to have a lower FDR

than alternative methods.

- The OSCA package also implements the OREML method for estimating the proportion of trait variance associated with all methylation probes combined. A global analysis of this kind would be interesting in the context of Lewy body pathology.
- Please add QQ plots for each of the three EWAS (discovery, replication, meta-analysis).
- Sensitivity analyses based on M values are said to generate very similar results, but I could not see these results.

Covariates:

- Smoking is known to have a major effect on CpG DNA methylation, but the analyses appear not to be adjusted for this potential confounder. Are the findings robust to adjustment for predicted smoking exposure?
- Braak stages are shared across multiple diagnoses (iLBD, LBD, PD, DLB); is it possible that the findings reflect diagnosis rather than Braak stage? i.e., are the Braak stage associations robust to adjustment for diagnosis?
- Were different methods/references used to deconvolute cell type proportions (CTPs) in the discovery and replication samples, and/or did the analyses differ in which CTPs were adjusted for? If so, why? Did the authors consider using other brain methylation references that may provide more granular estimates of CTPs, such as the Luo et al (2019) single cell methylation data (<https://doi.org/10.1101/2019.12.11.873398>)?
- There is a large difference (i.e., ~10-fold) in PMD between the discovery and replication cohorts. How stable is DNA methylation post-mortem, and is it possible that non-replication may be due to confounding of PMD with cohort? Why did the replication analysis not adjust for PMD?

Replication:

- I accept that BH FDR is a reasonable means of multiple testing correction in the discovery EWAS, but why was there no correction for multiple testing when assessing the 24 CpG probes in the replication analysis? Does any single probe replicate if this is done?

Association between CpG methylation and preclinical Lewy body pathology:

- If I understand correctly, the neurologically healthy controls with (n=29) and without (n=73) iLBD used to assess the 4 CpG probes associated with Braak stage, were all included in the EWAS that identified the probes. So, it isn't surprising that one or more would be differentially methylated between these groups, given iLBD comprises individuals with Braak stage 1-5, and all CONTR were Braak stage 0. This analysis really needs to be performed in a sample independent of the data used to identify the 4 significant probes.
- Fig 3 shows the distribution of cg13986157 methylation by diagnostic group, but what does this look like by Braak stage?

Cg03318382 association: any 22q11.2 CNV carriers in the data that might contribute to this association?

mQTLs:

- The authors used the brain xQTL data to screen for mQTLs for associated probes, but there are alternatives that would be worth exploring (e.g. Qi et al 2018, Nature Comms 9: 2282).
- If this resource revealed an mQTL for one or more significant probes, then the Summary-data-based Mendelian Randomization (SMR) method (Zhu et al. 2016 Nat Genetics) could be used to formally test for association of DNA methylation with the putative target genes (i.e., as opposed to the qualitative / annotation-based results presented currently).

Figures 1: please include details of the regression line (or correlation) in the captions. Can you add SEs to the beta estimates for each probe?

Figure 2: As above, add details of the regression line.

Response to reviewers' comments - NCOMMS-21-41216-T

We are grateful for the opportunity to resubmit our manuscript, entitled “Epigenome-wide association study of human frontal cortex identifies differential methylation in Lewy body pathology”. As requested, we have made comprehensive revisions to the article, aiming to address all of the reviewers' concerns. We have reanalyzed the replication data set from raw data to ensure an identical bioinformatic pipeline, and we have added a number of additional analyses as requested by the reviewers. Overall, we believe our manuscript is now greatly improved and we hope you find it acceptable for publication in Nature Communications. A point-by-point response to each of the reviewers' comments is given in italics below.

Reviewer #1 (Remarks to the Author):

Summary

Pihlstrøm et al. generated DNA methylation (DNAm) data for a large cohort of 322 postmortem frontal cortex brain samples from iLB, PD, DLB and control subjects. They performed analysis on both discovery and replication datasets to nominate 4 significant CpGs. Given that few DNAm studies have been performed in large cohorts of PD samples so far, the study made a significant contribution to the field by generating a valuable resource. The Discussion section is well written. My major concern for the work is that some of the analytical methods used for analyses might not be appropriate. In particular, there is a lack of control for multiple comparison adjustments in the replication study. There was also a lack of discussion on why different models were used in discovery and replication datasets, as well as a lack of consideration for differentially methylated regions.

We thank the reviewer for the positive comments about the work and writing. We respond point by point to the concerns and comments below.

Major Comments

1. Appropriateness of linear models. In the discovery cohort, there are different groups of samples from subjects with iLBD, PD, and DLB, for which the association between DNAm and Braak stage can be heterogeneous. The linear model should also adjust for heterogeneities introduced by these different groups, in addition to age, sex and proportions of neurons, to avoid potential confounding effects.

We thank the reviewer for this comment. A similar concern was also raised by Reviewer #2, so we acknowledge that we have not conveyed clearly how we mean to relate diagnosis to neuropathology in the model. Given that Lewy body pathology is the pathological hallmark of PD and DLB, Braak stage and diagnosis are necessarily highly correlated, as shown in Supplementary Figure 1. Note for instance that the control group is identical to Braak stage 0, and consequently does not contribute at all in an analysis adjusting for diagnosis.

*It has therefore never been our intention to separate the two and study their **independent** association with DNA methylation. The primary analysis is designed to assess association between DNA methylation and Lewy body stage as a quantitative outcome. This is analogous to the approach taken in recent large-scale EWAS studies of Alzheimer's disease pathology, which have typically used Tau Braak stage as a quantitative outcome, without adjusting for any clinical or pathological diagnosis (see meta-analysis of AD pathology studies based on this approach in Smith et al. Nat Commun 2021:12;3517).*

*We still believe it is important that the diagnostic groups are reported to give the reader a full impression of the sample set demographics. Furthermore, as the reviewers point out, any differential methylation we detect in our model could potentially be driven both by differences **across** and*

differences *within* diagnostic groups. We would expect that the strongest association signals will have elements of both. This is indeed what we observe when we include diagnostic group in the statistical model to assess Braak stage association within groups. The top probes from the primary analysis show a far weaker signal, yet still in the same direction and with $p < 0.05$.

In the revised manuscript, we have included equations to specify the regression models used. We present more clearly our rationale for choosing Braak stage as a quantitative variable in the main analysis. We also report the effect of adjusting for diagnosis on association statistics for the top probes, and we highlight the inability to disentangle diagnosis from neuropathology as a limitation in the discussion section.

2. Adjustment for cell-type proportions. Why were two different methods used for the analysis of discovery and replication cohort? For the discovery cohort, “We estimated the proportion of NeuN positive cells in each sample using reference data from flow-sorted frontal cortex cell populations as implemented in the minfi package.”

For replication cohort, “Cell proportions were derived using an algorithm which classifies the cellular populations in the cortex into three proportions (neuronal enriched, oligodendrocyte enriched and microglial enriched) (Shireby et al., manuscript in preparation).” Why not use the same method for both datasets? especially given the second one is not yet published.

We reply to points 2 and 3 jointly below.

3. Why was the discovery and replication cohort analyzed using different models?

For discovery, “We used a published algorithm and reference data to estimate the proportion of NeuN positive neurons in each tissue sample and included this variable as a covariate, as well as sex, age, postmortem interval, experiment plate, and the first three surrogate variables.”

But for the replication cohort, “linear regression models against Braak Lewy body stages were run controlling for age, sex, experiment plate and derived cellular proportions”

Why weren't postmortem interval and SVs used for replication cohort?

We thank the reviewer for raising points 2 and 3, which were also highlighted by Reviewer #2. The discrepancies in analysis pipeline resulted from minor differences in the standard workflows of collaborating labs. Although these differences are unlikely to have a major impact on the analysis results, we fully agree that the two datasets should be processed in an identical manner. We have now therefore exchanged raw data and reprocessed the replication data using the pipeline implemented in the analysis of the discovery data. Corresponding changes have been made to the Methods and Results sections.

Note that while the results are highly consistent with those previously reported, the inclusion of different covariates in the model results in slight adjustments to the resulting test statistics, causing a couple of changes to which probes fall on either side of $p < 0.05$ in the replication analysis. The revised manuscript now incorporates these updated results.

4. Another major concern is the lack of correction for multiple comparisons. In the analysis of the replication cohort, 24 probes from the discovery cohort were studied, FDR or Bonferroni correction should be made.

While we agree that comparison for multiple testing in both stages would have been ideal this is a highly conservative approach that has not generally been considered standard for two-stage discovery and replication studies e.g. in the GWAS literature, where sometimes even a one-sided $p < 0.05$ has been used (see for instance the last PD GWAS with a classical two-stage design: Nalls et al. Nat Genet 2014;46:989-993). More recent GWAS are often presented primarily as meta-analyses, with forest plots or leave-one-out analyses to demonstrate consistency across subcohorts.

We agree that when only 4 out of 24 probes nominated from the discovery stage replicate, one could be concerned about the robustness of these replications. However, the strong concordance of effect sizes between datasets (shown in Figure 2) demonstrates a highly significant concordance in direction of effect (21 out of 24 probes showing same direction effect, binomial sign test $p=0.00028$). This strongly indicates that the replication analysis is indeed reproducing the overall pattern of differential methylation, and justifies, in our opinion, the $p<0.05$ significance threshold for highlighting single probes as replicating.

Furthermore, we acknowledge that our replication dataset has limited statistical power with only 48 cases of clinical Lewy body disease. However, we still consider it a major strength of our study that we present previously unpublished results on a phenotype novel to the EWAS field from two independent cohorts analyzed with identical methodology. It would have been possible to primarily meta-analyze or split samples differently in order to achieve some probes passing adjusted thresholds twice, but we believe the current way to report the data is the most fair, robust and transparent approach.

We have revised the text to explain our considerations regarding significance levels in the replication stage and acknowledge as an important limitation that replication signals do not pass overall adjustment for multiple testing.

5. In the section “Association between CpG methylation and preclinical Lewy body pathology”, there was no multiple comparison correction for the 4 significant CpGs that were compared between neurologically healthy donors with iLBD (N = 29) and without iLBD (N = 73).

Please see our response to Reviewer #2 below. We agree that as this association of DNA methylation with neuropathological diagnosis is not independent of the primary Braak stage association, it may be misleading to present it as an additional result, and we have chosen to exclude this analysis from the manuscript.

6. Please also provide FDRs associated with meta-analysis P-values in Supplementary Table 2.

Good suggestion, we have included FDRs as requested.

7. I checked on the GitHub repository for scripts, but it is empty. The scripts should be made available before publication to allow assessment by reviewers on the reproducibility of the work.

We sincerely apologize for not having made the relevant repositories public before the first round of review. This has now been corrected and all scripts are accessible on GitHub at <https://github.com/lpihlstrom/projects>.

8. I recommend changing supplementary files to excel format. It's much easier for other researchers to analyze and re-use the data presented in excel files than a pdf file.

We thank the reviewer for this suggestion. We have submitted the supplementary files as Excel files this time.

9. The authors can also perform DMR analysis to detect differentially methylated regions in the datasets.

Thank you for raising this point. In our experience, DNA methylation array data are not optimal for DMR analysis, lacking the dense coverage of bisulfite sequencing data. Even more than for single CpG associations, the results are sensitive to the choice of analysis tools and parameters, and there is no standard methodology for replication of candidate DMRs in a smaller dataset.

The reviewer's comment prompted us to analyze DMRs using two different tools: DMRcate and ipdmr (from the ENmix package), which outputs 3 and 197 DMRs respectively in the discovery dataset, of which 2 are overlapping. No DMR is detected in the underpowered replication dataset, however, and single probes within the 2 DMRs identified with both methods do not replicate at $p < 0.05$ in the BDR data. These inconsistencies between DMR calling methods reflects the limitations of current analytical approaches for regional analyses of DNA methylation.

For these reasons, we have chosen not to include the DMR analysis in the manuscript, but we are of course willing to reconsider and report the results outlined above if requested by the editor.

Minor comments

1. Please change age to age at death throughout the manuscript

Thank you, we agree that "age at death" is more appropriate.

2. For the quality control of DNAm datasets, did the authors look at bisulfite conversion rate?

Yes, we have included this in the Methods section.

3. In Supplementary Table 1, please clarify the difference between "DLB" and "LBD" in the legend.

Good point. Lewy body disorder (LBD) is an umbrella term covering both PD and DLB, as these are not distinguished in the BDR data. We have included text to clarify this.

4. For Supplementary Figure 1, I suggest using distinct colors to differentiate different stages

Thank you for this suggestion, we have changed the colors as requested.

5. Figure 1 needs a better description. It might be better to use "effect size" rather than "beta values" throughout, as "beta values" can be confused with "methylation beta values". The legend also needs to indicate the estimated correlation and P-value.

The reviewer is quite right that "beta values" can be ambiguous and confusing in the context of methylation analyses. We have changed the legend and included the correlation statistics as suggested.

6. In Supplementary Table 2 ";" should be "."

Thank you for pointing this out. The table has been revised accordingly and resubmitted in Excel format.

7. In the text throughout, please replace "*" with "×". For example, $p = 5.0 * 10^{-8}$ should be 5.0×10^{-8}
Thank you - we have made the suggested replacements.

8. The manuscript would benefit from editing from a writing service. For example, "Parkinson's disease (PD) and dementia with Lewy bodies (DLB) are closely related progressive disorders with no available causal therapy" – what does "causal therapy" mean?

We have changed “causal” to “disease-modifying” and done our best to improve the writing - including careful language review by authors with English as their native first language.

Reviewer #2 (Remarks to the Author):

In this manuscript, Pihlstrom and colleagues describe analysis of DNA methylation in Lewy body pathology using data from 541 post-mortem frontal cortex samples from the Netherlands Brain Bank (discovery) and Normal Aging Brain Collection, Amsterdam (replication). This is a large sample for a brain based EWAS and is a significant new dataset in relation to understanding epigenetic changes involved in PD, LBD and DLB. The authors report 4 novel replicating differentially methylated probes, and a further locus from meta-analysis of the discovery and replication samples. These are notable findings, but I have several queries about the methods and interpretation of results.

We thank the reviewer for this nice summary and the positive evaluation of the dataset and analyses we have generated.

Analysis method:

– Do I understand correctly that linear regression was used to analyse Braak stage as a continuously distributed response variable? Inclusion of an equation(s) summarizing the analysis approach would be helpful.

Yes, correct. We thank the reviewer for the suggestion to include equations in the manuscript. We have added this and agree that it conveys the method more clearly.

– Given that the distribution of Braak stage is skewed, and differs significantly between discovery and replication samples, did the authors compare their primary results to those based on analysis of ordinal (e.g., Braak 0/1 vs 2/3 vs 4-6) or binarized (e.g., Braak 0-2 to Braak 3-6) variables?

We thank the reviewer for the comment. We analyzed Braak stage as a quantitative variable in the primary analysis in order to capture as much of the relevant variation as possible and maximize statistical power. When Braak stage is binarized (0-2 vs 3-6), the results are indeed highly concordant with the same directions of effect, yet with generally slightly attenuated p-values. We have included this supplementary analysis in the manuscript and highlighted the limitation of the model.

– Linear regression with covariates can result in significant inflation in EWAS due to unobserved confounding. Acknowledging the merits of the SV analysis, it would be reassuring if the results were robust using the mixed-linear-model-based MOMENT (and/or MOA) methods implemented in the OSCA package (<https://doi.org/10.1186/s13059-019-1718-z>) which have been shown to have a lower FDR than alternative methods.

We agree that the models implemented in the OSCA package provide a powerful tool to eliminate unobserved confounding in this type of array data. It is also highly conservative, however, as it controls for genome-wide correlation structures that are potentially disease related. Most major EWAS of brain pathology to date have used a linear regression approach similar to our main analysis (see e.g. meta-EWAS of AD by Smith et al. Nat Commun 2021:12;3517). A very recent article on ALS reported results using both a traditional linear model and the OSCA MOA approach and found that only a minority of associations remained significant with MOA (Hop et al. Science Transl Med

2022:14;eabj0264, although there was a very strong concordance of effect sizes between the two approaches). We opted to use the same approach and include MOA as an alternative model. We find a highly similar pattern of top probes and their effect sizes using MOA (See Supplementary Figure 3a), but not unexpectedly, associations do not pass the FDR significance threshold in the discovery data using this method. We highlight this as a limitation, but we still regard the primary results from the linear model valid and worth presenting to the scientific community.

– The OSCA package also implements the OREML method for estimating the proportion of trait variance associated with all methylation probes combined. A global analysis of this kind would be interesting in the context of Lewy body pathology.

This is indeed a valuable feature of the OSCA tool package, and we are grateful for the suggestion. The estimated proportion has been included in the manuscript.

– Please add QQ plots for each of the three EWAS (discovery, replication, meta-analysis).

Thanks for the suggestion - we have included these plots as well as for the MOA analysis

– Sensitivity analyses based on M values are said to generate very similar results, but I could not see these results.

We have included statistics on the top probes for all alternative models in Supplementary Data, as well as full summary stats on GitHub page.

Covariates:

– Smoking is known to have a major effect on CpG DNA methylation, but the analyses appear not to be adjusted for this potential confounder. Are the findings robust to adjustment for predicted smoking exposure?

We thank the reviewer for the comment. This is indeed a good point, as smoking is negatively associated with PD in epidemiological studies. However, tools to predict smoking status based on DNA methylation have primarily been trained on whole blood data and have been shown to have very limited precision in brain data. We applied the EpiSmokEr algorithm to our data but found no association between smoking scores and diagnosis/Braak stage. We also looked up previously published brain-specific smoking-associated probes but found no trend towards association with Braak stage in our data. These analyses have been mentioned in the manuscript and the lack of data on smoking during life in our participants has been highlighted as a limitation.

– Braak stages are shared across multiple diagnoses (iLBD, LBD, PD, DLB); is it possible that the findings reflect diagnosis rather than Braak stage? i.e., are the Braak stage associations robust to adjustment for diagnosis?

A similar point was raised by Reviewer 1. Please see our discussion about adjustment for diagnosis above under “Reviewer 1 -Major point 1”.

– Were different methods/references used to deconvolute cell type proportions (CTPs) in the discovery and replication samples, and/or did the analyses differ in which CTPs were adjusted for? If so, why? Did the authors consider using other brain methylation references that may provide more granular estimates of CTPs, such as the Luo et al (2019) single cell methylation data (<https://doi.org/10.1101/2019.12.11.873398>)?

We thank the reviewer for this comment, which was also raised by reviewer 1. We have now reanalysed the replication data using an identical pipeline. Taking advantage of single-cell data to further improve the methods to estimate cell type proportions is indeed a promising prospect. However, the referenced paper does not include data that can be used directly for such purposes by existing tools, and developing such methodology would be a substantial scientific undertaking that we consider to be well beyond the scope of our presently submitted article.

– There is a large difference (i.e., ~10-fold) in PMD between the discovery and replication cohorts. How stable is DNA methylation post-mortem, and is it possible that non-replication may be due to confounding of PMD with cohort? Why did the replication analysis not adjust for PMD?

We thank the reviewer for pointing this out. PMD was not associated with any of methylation PCs 1-5 in our data, suggesting there is no large, general effect, but that does not rule out an effect on individual probes. We have highlighted the differences in PMD across groups and datasets in the Methods section and included a reference to a methodological study on the effect of postmortem delay on DNA methylation in brain. We have revised the replication analysis to follow an identical pipeline - this time including PMD as a covariate, which did not substantially affect the results.

Replication:

– I accept that BH FDR is a reasonable means of multiple testing correction in the discovery EWAS, but why was there no correction for multiple testing when assessing the 24 CpG probes in the replication analysis? Does any single probe replicate if this is done?

A similar point was raised by reviewer 1. Please see our in-depth discussion above under “Reviewer 1 - Major point 4”.

Association between CpG methylation and preclinical Lewy body pathology:

– If I understand correctly, the neurologically healthy controls with (n=29) and without (n=73) iLBD used to assess the 4 CpG probes associated with Braak stage, were all included in the EWAS that identified the probes. So, it isn't surprising that one or more would be differentially methylated between these groups, given iLBD comprises individuals with Braak stage 1-5, and all CONTR were Braak stage 0. This analysis really needs to be performed in a sample independent of the data used to identify the 4 significant probes.

– Fig 3 shows the distribution of cg13986157 methylation by diagnostic group, but what does this look like by Braak stage?

We thank the reviewer for raising this point. We agree that it could be misleading to nominate probes based on one analysis and then present additional analysis of the same probes in the same data when the statistical hypothesis is not independent. We have therefore chosen to remove this analysis and Figure 3 from the manuscript. (We have chosen to instead include a volcano plot as a new figure in the revised main manuscript.)

Cg03318382 association: any 22q11.2 CNV carriers in the data that might contribute to this association?

We thank the reviewer for the suggestion to look into this. We have used the genotype data to screen for runs of homozygosity, which could indicate either identity by descent or deletion. We found no evidence of 22q11.2 microdeletions in our data, which is not unexpected given the low frequency of this CNV.

mQTLs:

– The authors used the brain xQTL data to screen for mQTLs for associated probes, but there are alternatives that would be worth exploring (e.g., Qi et al 2018, Nature Comms 9: 2282).

We thank the reviewer for this suggestion. We screened for mQTLs using the suggested dataset from Qi et al as well but found no mQTLs for our top probes. The analysis is added to the Results.

– If this resource revealed an mQTL for one or more significant probes, then the Summary-data-based Mendelian Randomization (SMR) method (Zhu et al. 2016 Nat Genetics) could be used to formally test for association of DNA methylation with the putative target genes (i.e., as opposed to the qualitative / annotation-based results presented currently).

We agree that if any mQTLs had been identified, the SMR method could potentially have provided statistical evidence linking DMRs to regulation of specific genes, which would have been stronger than the current annotation-based approach. However, as no mQTLs emerged in our analysis, we were unfortunately not in a position to take advantage of this tool.

Figures 1: please include details of the regression line (or correlation) in the captions. Can you add SEs to the beta estimates for each probe?

We thank the reviewer for the suggestions. We have added SEs and included details on the correlation as requested.

Figure 2: As above, add details of the regression line.

We have added the details as requested.

Additional notes:

While we were preparing the revision, a paper was published on a smaller 450K methylation array data from postmortem cortex of 14 PD patients and 10 control donors (Kaut et al. Life 2022; 12: 502). This study had no genome-wide significant probes, yet highlighted hypomethylation in the CYP2E1 locus as a main finding - also observed previously by the same group in a small (N=12) 2012 study. We noted a similar pattern in the BDR dataset, and we considered it appropriate to include a paragraph on this in the Results and Discussion sections, respectively.

Raw data from the BDR cohort have been submitted to GEO. Submission is in process for the NBB data and the Data Availability Statement both the manuscript and the reporting summary will be updated with the appropriate accession number before publication. Full summary statistics have been uploaded to our public GitHub repository.

REVIEWER COMMENTS

Reviewer #1 (Remarks to the Author):

This is a substantially revised manuscript. Most of my comments have been addressed well, and I appreciated the authors' efforts in adding clarifications and making changes to improve the manuscript.

Major Comment

1. My main concern is the interpretation of the effect estimates in Table 1 (4th column). I'm confused by the text under Table 1, which stated, "Effect size estimate corresponds to % fold change per Braak stage". In Methods, under "Linear regression and mQTL analyses", it mentioned the model "Methylation beta ~ Braak Lewy body stage + sex + age at death + cell composition + experiment plate + postmortem interval + SV1-3." was used for both discovery and replication analysis.

If beta values were used as the outcome variable, the effect size estimate (i.e., regression estimate for Braak stage) then corresponds to changes in normalized beta values per one unit change in the Braak stage, but the estimates in Table 1 (around 0.4 - 0.5) seem to be much larger than what one would expect for changes in beta values per 1 unit change in Braak stage. Also, Figure 3 shows the change in methylation per Braak stage is much less than 0.5.

I wonder if this weird result is because the no intercept model was used. The 0 in the model.matrix () indicates a no intercept model was used; this may have forced beta value to be a specific value for samples with Braak stage = 0

```
aSyn_design_3sv <- model.matrix(~ 0 + Braak_aSyn_stage + Age_death + Sex + PMD_min +  
NeurProp + Plate + SV1 + SV2 + SV3, aSyn_spctr_pheno)
```

Please clarify how "Effect estimate" in Table 1 was computed.

Please also remove "NBB (%)" and "BDR (%)" in Table 1

Minor Comment

1. In Supp Figure 3 (Manhattan plots), please remove "Plots were generated using the R package qqman." in legend

2. In Methods, under "DNA methylation analyses, data normalization, and quality control", It was mentioned "The NBB experiment included technical replicates for 30 samples". Technical replicates are not independent samples. Only independent samples should be included in regression models. Please clarify.

Reviewer #2 (Remarks to the Author):

The authors have done an excellent job responding to the issues raised in my review. I have no further concerns.

Response to reviewers' comments - NCOMMS-21-41216A

We are grateful for the opportunity to resubmit our manuscript, entitled “Epigenome-wide association study of human frontal cortex identifies differential methylation in Lewy body pathology”. We appreciate that both reviewers acknowledge our efforts to address the concerns raised in the first round of reviews and find the manuscript substantially improved.

Please see our point-by-point response to the comments from Reviewer #1 in italics below, which we hope should clarify the remaining concerns. Please note that we have now completed submission to GEO for the NBB methylation data and accession number has been added to the data availability statement. The Reporting Summary and Editorial Policy Checklist have been updated accordingly.

We thank you for your consideration and sincerely hope that you will find our revised article acceptable for publication in Nature Communications.

Reviewer #1 (Remarks to the Author):

This is a substantially revised manuscript. Most of my comments have been addressed well, and I appreciated the authors' efforts in adding clarifications and making changes to improve the manuscript.

We thank Reviewer #1 for this positive evaluation.

Major Comment

1. My main concern is the interpretation of the effect estimates in Table 1 (4th column). I'm confused by the text under Table 1, which stated, “Effect size estimate corresponds to % fold change per Braak stage”. In Methods, under “Linear regression and mQTL analyses”, it mentioned the model “Methylation beta ~ Braak Lewy body stage + sex + age at death + cell composition + experiment plate + postmortem interval + SV1-3.” was used for both discovery and replication analysis.

If beta values were used as the outcome variable, the effect size estimate (i.e., regression estimate for Braak stage) then corresponds to changes in normalized beta values per one unit change in the Braak stage, but the estimates in Table 1 (around 0.4 – 0.5) seem to be much larger than what one would expect for changes in beta values per 1 unit change in Braak stage. Also, Figure 3 shows the change in methylation per Braak stage is much less than 0.5.

I wonder if this weird result is because the no intercept model was used. The 0 in the model.matrix () indicates a no intercept model was used; this may have forced beta value to be a specific value for samples with Braak stage = 0

```
aSyn_design_3sv <- model.matrix(~ 0 + Braak_aSyn_stage + Age_death + Sex + PMD_min +  
NeurProp + Plate + SV1 + SV2 + SV3, aSyn_spctr_pheno)
```

Please clarify how “Effect estimate” in Table 1 was computed.

Please also remove “NBB (%)” and “BDR (%)” in Table 1

*We understand that we may have confused the reviewer and acknowledge that this important detail was not made sufficiently clear in the manuscript. The numbers presented were coefficients and SEs from linear regression (logFC from limma::topTable) **multiplied by 100**, We chose this because we were concerned that very small numbers such as effect size 0.0042, SE 0.00011 were a bit hard to read and interpret, and didn't look too good in a table. The intention was to then refer to beta values*

on a %-scale from 0 to 100, rather than the standard 0-1. This is also why we stated “% fold change” in the text under the table and included “%” in the column heading, as the reviewer suggested be removed.

On second thoughts, seeing how this appeared confusing, we now instead present the raw coefficients throughout in both text, figure and tables. In the Table 1 legend, the effect size estimate is now specified as follows: “Effect estimates are coefficients from linear regression, corresponding to the change in methylation beta value for each unit of increase in Braak stage”. We have removed “%” from the column heading of Table 1.

Some authors also choose to report the effect size contrasting the two extremes, Braak 0 versus Braak 6 in this case, which would mean multiplying the numbers by 6. However, we suggest to leave it like this, unless the editor should have other preferences.

Just to be sure, we repeated analysis without the “0” in the model to include an intercept as suggested by the reviewer. This made no difference at all to coefficients or p-values, but we can easily understand how the reviewer got confused and thought of such possible explanations.

Minor Comment

1. In Supp Figure 3 (Manhattan plots), please remove “Plots were generated using the R package qqman.” in legend

Thank you, we have removed this sentence as suggested.

2. In Methods, under “DNA methylation analyses, data normalization, and quality control”, It was mentioned “The NBB experiment included technical replicates for 30 samples”. Technical replicates are not independent samples. Only independent samples should be included in regression models. Please clarify.

Yes, of course these technical replicates were used for quality checks only and thus removed before any statistical hypotheses testing and not counted among the 322 NBB samples. In order to clarify, we have added the following statement in the Methods section: “The NBB experiment included technical replicates for 30 samples, which were used for quality checks only and removed before statistical hypothesis testing.”

Reviewer #2 (Remarks to the Author):

The authors have done an excellent job responding to the issues raised in my review. I have no further concerns.

We are very grateful for this positive evaluation from Reviewer #2.

REVIEWERS' COMMENTS

Reviewer #1 (Remarks to the Author):

The authors addressed my previous concerns well and I don't have any additional concerns.

Response to reviewers' comments - NCOMMS-21-41216B

We are grateful for the opportunity to submit a final version of our manuscript, entitled “Epigenome-wide association study of human frontal cortex identifies differential methylation in Lewy body pathology”. We appreciate the efforts of the reviewers in evaluating and helping to improve the manuscript and we are happy to see that there are no additional concerns.

Reviewer #1 (Remarks to the Author):

The authors addressed my previous concerns well and I don't have any additional concerns.

We thank Reviewer #1 for this positive evaluation.